# Talimogene Laherparepvec (T-VEC): An Intralesional Cancer Immunotherapy for Advanced Melanoma

**DOI:** 10.3390/cancers13061383

**Published:** 2021-03-18

**Authors:** Pier Francesco Ferrucci, Laura Pala, Fabio Conforti, Emilia Cocorocchio

**Affiliations:** 1Tumor Biotherapy Unit, Department of Experimental Oncology, European Institute of Oncology, IRCCS, 20141 Milan, Italy; 2Division of Melanoma, Sarcoma and Rare Tumors, European Institute of Oncology, IRCCS, 20141 Milan, Italy; laura.pala@ieo.it (L.P.); fabio.conforti@ieo.it (F.C.); 3Hemato-Oncology Division, European Institute of Oncology, IRCCS, 20141 Milan, Italy; emilia.cocorocchio@ieo.it

**Keywords:** oncolytic virus, talimogene laherparepvec, T-VEC, GM-CSF, intratumoral immunotherapy, melanoma

## Abstract

**Simple Summary:**

Talimogene laherparepvec (T-VEC; IMLYGIC®, Amgen Inc.) is the first oncolytic viral immunotherapy to be approved for the local treatment of unresectable metastatic stage IIIB/C–IVM1a melanoma. Its direct intratumoral injection aim to trigger local and systemic immunologic responses leading to tumor cell lysis, followed by release of tumor-derived antigens and subsequent activation of tumor-specific effector T-cells. Its approval has fueled the interest to study its possible sinergy with other immunotherapeutics in preclinical models as well as in clinical contextes. In fact, it has been shown that intratumoral administration of this immunostimulatory agent successfully synergizes with immune checkpoint inhibitors. The objectives of this review are to resume the current state of the art of T-VEC treatment when used in monotherapy or in combination with immune checkpoint inhibitors, describing the strong rationale of its development, the adverse events of interest and the clinical outcome in selected patient’s populations.

**Abstract:**

Direct intralesional injection of specific or even generic agents, has been proposed over the years as cancer immunotherapy, in order to treat cutaneous or subcutaneous metastasis. Such treatments usually induce an effective control of disease in injected lesions, but only occasionally were able to demonstrate a systemic abscopal effect on distant metastases. The usual availability of tissue for basic and translational research is a plus in utilizing this approach, which has been used in primis for the treatment of locally advanced melanoma. Melanoma is an immunogenic tumor that could often spread superficially causing in-transit metastasis and involving draining lymph nodes, being an interesting model to study new drugs with different modality of administration from normal available routes. Talimogene laherperepvec (T-VEC) is an injectable modified oncolytic herpes virus being developed for intratumoral injection, that produces granulocyte-macrophage colony-stimulating factor (GM-CSF) and enhances local and systemic antitumor immune responses. After infection, selected viral replication happens in tumor cells leading to tumor cell lysis and activating a specific T-cell driven immune response. For this reason, a probable synergistic effect with immune checkpoints inhibition have been described. Pre-clinical studies in melanoma confirmed that T-VEC preferentially infects melanoma cells and exerts its antitumor activity through directly mediating cell death and by augmenting local and even distant immune responses. T-VEC has been assessed in monotherapy in Phase II and III clinical trials demonstrating a tolerable side-effect profile, a promising efficacy in both injected and uninjected lesions, but a mild effect at a systemic level. In fact, despite improved local disease control and a trend toward superior overall survival in respect to the comparator GM-CSF (which was injected subcutaneously daily for two weeks), responses as a single agent therapy have been uncommon in patients with visceral metastases. For this reason, T-VEC is currently being evaluated in combinations with other immune checkpoint inhibitors such as ipilimumab and pembrolizumab, with interesting confirmation of activity even systemically.

## 1. Introduction

Melanoma is an aggressive cutaneous malignancy that is growing in incidence worldwide [1]. Local or regional disease may be cured with surgical treatment in many patients, but relapse is common, especially in those presenting with high risk features such as ulceration, high mitotic cell numbers and elevated thickness of the primary tumor [2]. When draining lymph nodes are involved, risk of relapse is even higher and could be reduced by adjuvant targeted or immune-therapies [3,4,5,6]. Due to the specific tropism to the skin, in-transit metastases often occur by spreading of melanoma cells through the dermal lymphatics and presenting as cutaneous or subcutaneous lesions, generally between the primary tumor site and its regional lymphatic basin [7]. On the other hand, melanoma may metastasize to any sites, disseminating to the skin and soft tissues distant from the primary or even systemically. In both of these clinical scenarios, surgery is not considered the optimal first line option, due to the extent of disease and high risk of reaching no radicality. However, isolated limb perfusion [8], electrochemotherapy [9], and other locoregional or topical therapies [10], have also been experimented when visceral disease is not present, but usually with only a transitory and limited local disease control good for symptoms palliation. 

In general, different kinds of cytokine-based intralesional therapies, like PV-10 Rose Bengal, Bacillus Calmette Guerin (BCG), interleukin-2 (IL-2), and interferon alpha (IF-alfa), have been used to control locoregional disease [11,12,13,14,15]. PV-10 and BCG were able to induce mainly a regression of most of the injected melanoma lesions, while the others also demonstrated activity in non-injected ones, showing a sort of “bystander effect” in up to half of the treated patients [16,17]. However, all these agents may be mostly useful in highly selected cases for local disease control, since their cutaneous toxicity and the lack of a durable benefit have limited their clinical utility. 

Recently, other intratumoral immunotherapies with more elaborated mechanisms of action and rationale of use, demonstrated promising antitumor activity with tolerable toxicities on both local and systemic disease. They include non-oncolytic viral therapies (toll-like receptor agonists) and oncolytic viral therapies (CAVATAK, HF10) [18,19,20].

In this scenario, T-VEC, the first oncolytic herpes virus approved by the U.S. Food and Drug Administration for the treatment of unresectable melanoma recurrent after initial surgery, was designed to enhance local and systemic antitumor immune responses and studied as a monotherapy or in combination with possibly synergistic drugs [21]. Results obtained in recently published clinical trials, showed an effective efficacy of this intralesional treatment on both injected and not-injected lesions, low collateral effects and a significant systemic disease control especially through the combination with immune checkpoint inhibitors.

## 2. T-VEC Mechanisms of Action and Preclinical Activity

Oncolytic viruses include wild-type and modified live viruses and represent an innovative approach to cancer immunotherapy. In particular, T-VEC is a first-in class oncolytic virus, intra-lesionally delivered, constituted by a genetically engineered herpes simplex virus type 1 (HSV-1) that selectively replicates in tumor tissue, lyses tumor cells, while promoting anti-tumor immunity [22]. Moreover, human *GM-CSF* gene is inserted in the virus and this results in local GM-CSF production, able to enhance the influx and activation of dendritic cells (DCs) needed to start a systemic antitumor response. Finally, the genes encoding neurovirulence factor ICP34.5 and ICP47 are functionally deleted in the viral drug [23,24]. Functional deletion of ICP34.5 attenuates viral pathogenicity and allows the virus to replicate selectively in tumors, while deletion of the ICP47 gene reduces virally mediated suppression of antigen presentation and increase the expression of the HSV US11 gene, which in turn, promotes virus growth in tumor cells without impairing tumor selectivity (Table 1).

In general, cancer immunotherapy relies on the recognition of tumor-associated antigens by immune cells and involve innate immunity components, as well as proinflammatory cytokines like tumor necrosis factor α and interferon-γ [25,26,27,28]. When T-VEC immunotherapy is given intratumorally, it causes direct tumoral cells to die and acts as a source of antigens, favoring local recruitment of immune cells into the tumor microenvironment. Subsequently primed T cells induce a systemic polyclonal antitumor response, which can potentially address intra- and intertumoral heterogeneity [29,30,31]. In particular, T-VEC, entering the cancer cells through the herpes virus glycoproteins on the cell surface, favorably replicates and ultimately triggers cell lysis [32]. In this way, tumor-derived antigens (TDA), GM-CSF and new viral particles are released allowing just one-time infection of surrounding tumoral cells and enhancing TDA spreading [33]. In addition, danger-associated molecular pathways are activated, evoking a systemic antitumor effect and a complete immune response to occur (Figure 1) [32].

Practically, T-VEC is administered through a series of intralesional injections and bypasses absorption barriers, so traditional pharmacokinetic studies were not relevant and therefore not performed. On mouse models, the highest concentration of viral DNA is detectable in the site of injection. However, viral DNA was also found at lower concentrations in non-target sites, like blood, spleen, lymph node, liver, heart and kidney, while it was absent in the bone marrow, eyes, lachrymal glands, nasal mucosa and feces [34,35]. The attenuated virus is eliminated via host-defense mechanisms, including autophagy and adaptive immune responses, but viral DNA may persist in neuronal cell bodies surrounding the site of injection without clinical implications [34,35].

Andtbacka et al. reported similar results on humans in a Phase II study which enrolled 20 patients with melanoma: T-VEC’s DNA was present in 85% of their blood samples. Vital viruses were also documented at the intralesional site and, interestingly, viruses were able to maintain their infectivity capacity in around 15% of patients within the first week after initial injection. On the other hand, transmission of T-VEC from patients to hospital staff or family members was not reported with proper administration and handling process [36].

## 3. T-VEC Clinical Experience in Monotherapy

T-VEC had been firstly tested in a Phase I study, where it demonstrated biological activity in terms of active viral replication, local reactions (erythema and inflammation of the injected lesions), granulocyte-macrophage colony stimulating factor (GMCSF) expression, tumor necrosis and apoptosis, as evaluated in post-treatment biopsies. An acceptable toxicity profile, mainly low grade and characterized by fever, myalgia, chills and local reactions, was reported [37].

In a subsequent single arm Phase II study, conducted on 50 patients with locally advanced or metastatic melanoma, T-VEC obtained 26% of overall response rate (ORR), with 8 complete responses (CR) and 5 partial responses (PR). Shrinking in the tumor burden of injected and non-injected lesions, included visceral lesions, was reported, with most of the responses (92%) maintained for almost 3 years. Overall survival (OS) was 58% at 12 and 52% at 24 months, respectively [38].

Another report by Kaufman et al., showed an increase in MART-1 CD8+ T cells and IFN-gamma produced by tumor infiltrating lymphocytes (TILs) in injected lesions undergoing regression after vaccination compared with T-cells derived from lesions of untreated melanoma patients. A significant decrease in CD4+ FoxP3+ regulatory T-cells was also reported in injected lesions compared with those non-injected, both from the same patient or from other untreated melanoma patients. Similarly, myeloid derived stem cells (MDSCs) were significantly decreased in melanoma lesions treated with T-VEC compared with not treated ones or with not vaccinated patients. These data support the evidence that T-VEC is able to induce a valid local and systemic T-cell immunity in melanoma patients with advanced disease [39].

The Phase III OPTIM study enrolled 436 patients with locally advanced or metastatic melanoma, randomized to receive intra-lesionally T-VEC or subcutaneous GM-CSF with a two to one ratio. They had at least one cutaneous, subcutaneous or nodal lesion and no more than 3 visceral lesions. Prognostic characteristics were well balanced between the two arms: 163 patients (55%) in the T-VEC arm and 86 patients (61%) in the GM-CSF arm had stage IIIB–IVM1a disease, while the remaining had a more advanced melanoma (stage IVM1b/c); most of them (90%) presented low levels of LDH at the moment of the enrollment. Half of the patients did not receive any prior therapy, while mutational status was known in only 30% of patients, half of which had a BRAF mutated melanoma [40].

The first dose of T-VEC was administered at 10^6^ pfu/mL (dose showed to seroconvert HSV-seronegative patients, in the previous Phase I study). Subsequent doses were administered at 10^8^ pfu/mL three weeks after the first dose and then once every two weeks, for a total dose of 4 mL per dose each time. GM-CSF 125 g/m^2^ was administered subcutaneously once daily for 14 days in 28-day cycles. The primary endpoint was durable response rate (objective response lasting continuously for at least 6 months or more), while OS and overall response rate (ORR) were secondary endpoints [37,40]. Median (range) duration of treatment was 23.1 weeks (0.1–176.7) in the T-VEC arm and 10.0 weeks (0.6–120.0) in the GM-CSF arm. Median follow-up (time from random assignment to analysis) in the final analysis of OS was 49 months.

In the final analysis of this study, Andtbacka and colleagues reported significantly higher durable response rate (DRR) with T-VEC (19.3%) than GM-CSF (1.4%) as per investigator assessment (unadjusted odds ratio, 16.6; 95% CI, 4.0–69.2; *p* <  0.0001). Overall, 50 (16.9%) and 1 (0.7%) patients in the T-VEC and GM-CSF arms, respectively, achieved CR, while 43 (14.6%) and 8 (5.7%) achieved PR with a disease control rate of 76.3% versus 56.7% with T-VEC and GM-CSF, respectively. Median OS was also in favor of T-VEC arm, reaching 23.3 (95% CI, 19.5–29.6) months versus 18.9 months with GM-CSF (95% CI, 16.0–23.7). Consequently, reduction in the risk of death was 21% with T-VEC versus GM-CSF. Subgroup analyses showed enhanced T-VEC efficacy in those patients with stage IIIB to IV M1a (28.8% DRR, 46.0% ORR and 79.1% DCR) and in a first line setting in respect to more advanced diseases. Finally, T-VEC toxicity profile was acceptable, with most common adverse events (AEs) represented by flu-like syndrome symptoms, as fatigue, chills, and pyrexia. The only grade 3 or 4 AE occurring in <2% of T-VEC–treated patients was cellulitis and there were no treatment-related deaths [40].

However, the Phase III trial was hardly criticized because of the choice of a comparison arm which was not considered the correct standard of treatment, even though it was clearly designed in a period when new drugs and combinations were not approved yet (Table 2).

## 4. T-VEC Clinical Experience in Combination

The evidence of local and systemic immune responses induced by T-VEC alone, supported the rationale for designing trials evaluating the efficacy of its combination with other immunotherapies, in particular immune checkpoint inhibitors.

Preclinical studies in melanoma murine models with the combination of intralesional T-VEC and anti-cytotoxic T-lymphocyte antigen 4 (antiCTLA-4) monoclonal antibodies were conducted to better characterize local and systemic antitumor immune responses driving efficacy [41,42].

In the mouse, the treatment with antiCTLA-4 (ipilimumab) and T-VEC was shown to be able to cure all injected tumors and around half of non-injected tumors. A significant increase in T-cells (CD3^+^/CD8^+^) was observed in injected and contralateral tumors at one week since the administration and ex vivo analyses showed these cytotoxic T-lymphocytes were tumor-specific. Increased neutrophils, monocytes and chemokines were observed in injected tumors only, while depletion of CD8^+^ T-cells with anti-CD8 antibodies treatment was able to abolish all systemic efficacy and significantly decrease local efficacy, too. Finally, T-VEC and checkpoint blockade combination resulted in increased tumor-specific CD8^+^ anti-AH1 T cells and higher systemic efficacy [43].

Subsequently, a randomized Phase Ib/II study was planned to evaluate the combination of T-VEC with Ipilimumab versus Ipilimumab alone in unresectable stage IIIB-IIIC and IV melanoma patients. One hundred ninety-eight patients were enrolled and the ORR was significantly improved in the combination arm versus the ipilimumab one (*p* = 0.033). Thirty-eight patients in the combination arm [39% (CR, 13%; PR, 26%)] and 18 patients in the ipilimumab arm [18% (CR, 7%; PR, 11%)] had an objective response. Even in this case, effectiveness in the combination arm was higher for patients with low tumor burden (stage IIIB/IIIC/IVM1a) in comparison to high tumor burden (IVM1b and IVM1c) showing an ORR of 44% and 33%, respectively [44].

Chesney and colleagues reported also specific data on subgroup analysis based on BRAF molecular status. In particular, although response rates were quite high in patients with both BRAF wild-type and BRAF mutant tumors, it is interesting to note, that the magnitude of efficacy favoring the combination treatment was greater in the first subgroup, (42% vs. 10%). The ORR was also greater in the combination arm for patients with BRAF wild-type versus BRAF mutant tumors (42% vs. 34%), respectively. The median time to response was 5.8 months in the T-VEC/ipilimumab arm (*n* = 38) and not estimable in the ipilimumab arm (*n* = 18), however, the median DRR was not reached in either arms. Finally, the median progression free survival (PFS) was 8.2 months in the duplet arm and 6.4 months in the monotherapy arm (*p* = 0.35) [44].

T-VEC toxicity was consistent to monotherapy studies, with the flu-like symptoms as the most characteristic complication. Grade 3 ipilimumab-related toxicities occurred similarly in both arms with no additive effect. Though, combination treatment was not associated with unexpected AEs or increase in toxicity, suggesting that it is tolerable in patients with advanced melanoma. Although this trial was positive, it has some limitations, mainly represented by the relatively small number of patients and the short follow up at the time of analysis [44]. 

However, other combination strategies are being tested. The MASTERKEY-265 trial, a Phase Ib/III study, is evaluating the administration of T-VEC with the anti-programmed death-1 antibody (antiPD1) Pembrolizumab in previously untreated patients with stage IIIB to IV melanoma. Results of the Phase Ib, enrolling 21 patients, showed that the combination of T-VEC with Pembrolizumab was associated with clinical benefit. At a median follow up of 18 months, the confirmed ORR was 61.9% (95% CI, 38.4–81.9%), reaching a very high CR rate of 33.3%. The combination treatment resulted in a more than 50% size reduction in 82% of injected, 43% of non-injected non-visceral, and 33% of non-injected visceral lesions [45]. Median PFS and OS were not reached at the time of last follow up.

Interestingly, at the subsequent data cutoff performed on 11 June 2018, all 21 enrolled patients were off treatment: six were dead and 15 still in long-term follow-up. With a median follow-up time of 36.8 months, two patients had their tumor response changed from previous partial response/stable disease to CR. Therefore, ORR improved to 67% (14/21 pts) with a CR rate reaching the incredible value of 43% (9/21 pts). When looking among the 14 responders, 12 (57%) remain in response, including the nine with CR. Median PFS and OS have still not being reached at the data cutoff, however, 36-mo PFS and OS rates were 53.6% and 71%, respectively. Importantly, no additional safety signals were detected [46].

The randomized, double blind, Phase III part of this study is now ongoing (ClinicalTrials.gov: NCT02263508) and plans to accrue 660 patients, half receiving the combination therapy and half receiving pembrolizumab with intratumoral placebo in the control arm [47].

Interestingly, to further evaluate T-VEC systemic effects, a Phase II single arm study (ClinicalTrials.gov: NCT02366195) for the evaluation of biological biomarkers is also ongoing and will evaluate baseline and post T-VEC reactions in injected and not-injected tumors from more than 100 patients [48]. This study will provide important data on tumoral microenvironment of non-injected metastases and could hopefully led to the approval of a new combination therapy at least as a second line of treatment [49].

Finally, another trial which is ongoing is a Phase II study (ClinicalTrials.gov: NCT02211131) for efficacy and safety of T-VEC plus surgery versus surgery in the neoadjuvant setting in stages IIIb-IIIc or IV M1a melanoma patients (Table 2) [50].

## 5. Real-World Experience with T-VEC

Few retrospective reports are available on real-world experience with T-VEC in monotherapy (none in combination), mainly due to the fact that this drug is currently approved, and though available, in countries like United States, Australia, UK, Germany, Switzerland and France, while there is no expanded access programs or compassionate use open so far. As a matter of fact, all the other countries could use T-VEC only within clinical trials.

The most extended real-world experience with T-VEC monotherapy was published by Louie et al. as a multi-institutional report on 80 patients with stage IIIB–IV melanoma [51]. After 9 months’ median follow-up, patients were evaluated for locoregional response, showing 31 (39%) CR and 14 (18%) PR, which resulted in an ORR of 57%. AEs were generally mild, with the most common flu-like symptoms seen in 22 (28%) patients [51].

Franke et al. reported a single-institution experience with T-VEC monotherapy on 26 patients with stage IIIB–IVM1a melanoma, using a treatment schedule consistent with the OPTiM study protocol [52]. Among the 26 patients enrolled, most where in first line, while 3 (11.5%) underwent prior systemic treatment, with BRAF and/or MEK inhibitors, ipilimumab plus nivolumab combination or in a Phase I trial (3.8%). With a median follow-up time of 12.5 months, 16 (61.5%) patients had a CR and 7 (26.9%) a PR, resulting in an impressive ORR of 88.5% and disease control rate (DCR) of 92.3%. No new signs of toxicity were reported compared with those observed in the OPTiM trial [52].

Another single-institution experience was reported by Perez et al. and included 27 patients: five patients had an extended disease with visceral involvement (IV M1c melanoma), 14 (52%) had received no prior treatment and five (19%) had been treated with immunotherapies. With a median follow-up time of 8.6 months, 23 patients were evaluated for response: 10 patients (43.5%) achieved a CR and 3 (13.1%) a PR, with an ORR of 56.5% and a DCR of 78.3%. Also in this case, AEs were consistent with previous reports [53].

Comparing these data with those achieved in the Phase II and OPTIM clinical trials, they appear much better at least in terms of locoregional responses, giving a strong evidence supporting the use of T-VEC even in second line alone or in combination with checkpoint inhibition. On the other hand, fewer clear indications exist on duration and extension of general disease control on visceral/distant metastases.

However, evaluation of retrospective data should be interpreted with caution, because of the small number of analyzed patients and frequent differences in tumor assessments choices. In fact, looking more in detail on patient’s characteristics, clinical trials included a greater number of Stage IV disease patients and subgroup analyses, as performed in the OPTIM trial, showed that patients with an earlier-stage melanoma are more likely to respond compared with those with a more advanced disease [54,55].

An anecdotal case treated at the European Institute of Oncology, is reported in Figure 2.

## 6. Translational Research

T-VEC may be particularly useful in conditions of primary or acquired resistance to immunotherapy, a setting where systemic immunization cannot sustain cytotoxic T-cells effector functions, because it is overcome by the highly immunosuppressive tumor microenvironment. However, limited biomarker studies have been reported that examined the immune modulation effects occurring after T-VEC treatment in patients.

Resistance by local immunosuppression has been shown to be multifactorial and caused by the recruitment of inhibitory cells [regulatory T cells (Tregs) and suppressive myeloid-type cells], the release of inhibitory soluble factors [IL-10, transforming growth factor-beta (TGF-β), and vascular endothelial growth factor (VEGF)], and the expression of inhibitory cell surface receptors (FasL, IFN-γ, PD-L1 and B7-H1) [56,57,58,59,60,61,62]. Those events are coupled with immunoediting processes occurring at a systemic level and supports the importance of monitoring the local site of tumor growth in addition to peripheral circulation [61].

In a Phase 2 study of intralesional T-VEC in patients with Stage IIIC-IV melanoma, Kaufman and colleagues performed a T-cell phenotypic characterization on tumor infiltrating (TILs) and peripheral blood (PBLs) lymphocytes, obtained from T-VEC treated versus not treated patients [39]. In particular, authors showed a discrepancy between quantitative and qualitative immunophenotypes in TILs and PBLs. Firstly, characterizing those samples they showed the presence of Tregs in both peripheral blood of melanoma patients and, even at higher levels, in the tumor microenvironment of established melanoma lesions. Secondly, they found a profound immunophenotype difference in those Tregs present in non-vaccinated melanoma tumors compared with T-VEC-treated lesions, proposing this may be related to more activated T-cells within the latter. Thirdly, consistent with other reports, they also found Tregs and suppressor myeloid-like cells (able block T-cell proliferation in vitro) at higher concentrations in established melanoma lesions but at lower frequency in T-VEC-treated lesions. Finally, authors showed that T-VEC intratumoral treatment is able to induce local and systemic MART-1-specific CD8^+^ effector cells, supporting the achievement of a potent local and systemic T cell immunity in melanoma patients treated with this drug [39]. Unfortunately, this biomarker study was limited by the lack pre-treatment biopsies for intra-patient comparison, but was confirmed by further studies, especially those using combination treatments with checkpoint inhibitors.

In the T-VEC plus ipilimumab study, Puzanov et al. reported that both T- VEC alone and even more in combination with ipilimumab, were able to increase activated CD8^+^ T-cells from baseline levels, probably reverting the suppressing microenvironment [63]. In fact, an immune-suppressing or “cold” tumoral environment would be unlikely to respond to immunotherapy, and it is considered the major mechanism underlying the development of primary resistance. In particular, patients having low densities of CD8+ T-cells on baseline biopsies, usually lack of significant IFN-γ expression, which result in low PD-L1 expression and should probably be treated with combination strategies able to turn the tumor into a “hot” one and though more sensible to immunotherapy [45,64,65,66]. For this reason, to significantly increase the response rate to single-agent immune checkpoints by avoiding the development of resistance, combination therapies should address the possibility to change the microenvironment status into a more permissive one. The issue is to increase intratumoral infiltration by CD8+ T-cells, leading to their neoantigen’s recognition and tumor specificity strengthening, in order to being able to reverse the primary resistance to blockade therapy [45,67]. In particular, Ribas and colleagues sustain that talimogene laherparepvec may provide this combinatorial effect [45].

The authors report evidence that T-VEC local administration contributed to a systemic antitumor effect by the increase in circulating CD8^+^ and CD4^+^ T-cells and by the ability to increase inflammation in tumors after injection and before pembrolizumab treatment. Moreover, analyzing by immunohistochemistry those biopsies obtained from patients enrolled in the Phase Ib trial, they find an increased presence of CD8^+^ T cells in 75% of injected lesions and they observed that this increase in density was associated with response to combination therapy. Tumor gene expression analysis was also performed, demonstrating an elevation in CD8-α and IFN-ɤ mRNA after T-VEC administration. The level of PD-L1 expression was also increased, but clinical response did not appear to be associated with baseline PD-L1 status or CD8+ T-cell infiltration [45].

In a Phase II, single-arm, biomarker study of T-VEC monotherapy (ClinicalTrials.gov: NCT02366195), performed on biopsy samples obtained from uninjected lesions, also Gogas and colleagues reported that T-VEC treatment is able to increase CD8+ tumor-infiltrating lymphocytes, granzyme B+ effector CD8+ T cells, memory CD8+ T cells, and CD8+ T cells expressing checkpoint markers, but not macrophages [48,49]. Taken all together these immunological changes are converging and give a scientific support of the systemic effect induced by T-VEC especially when combined with checkpoint inhibitors. However, this topic remains an active area of investigation, and relevant available data are currently limited, but will hopefully become available in the near future thank to the ongoing clinical trials [68,69,70,71].

## 7. Conclusions

The landscape of options for advanced melanoma is rapidly improving and progressing, with drugs and combinations able to significantly expand patients’ window of expectations. However, adverse events occurring during treatment can limit the dose intensity and duration of those treatments, thus consequently affecting their effectiveness. Moreover, despite their tremendous impact on outcomes, most of the patients still relapse and die of their disease, leading to the need of expanding the therapeutic armamentarium with new drugs, sequences or combinations. Furthermore, with the advent of new technologies, therapeutic options are becoming more and more multidisciplinary, being surgery often avoided thanks to the adoption of new approaches requiring radiotherapy or interventional radiology expertise in order, for example, to reach visceral and deep internal lesions for intralesional injection of drugs.

T-VEC is the first genetically modified herpes simplex virus-1-based oncolytic immunotherapy approved by FDA and EMA for the treatment of unresectable, cutaneous, subcutaneous and nodal lesions in patients with melanoma recurrent after initial surgery [72,73]. This intratumoral injectable drug is designed to preferentially replicate in tumors, produce GMCSF, and stimulate antitumor immune responses both locally and systemically. As extensively discussed in this review, it has shown efficacy in monotherapy and in combination with immune checkpoints inhibitors.

Efficacy has been demonstrated mainly on injected lesion, but also an abscopal systemic effect was evident on metastasis in distant organs. Like most of the immunotherapies effectiveness is not immediately translatable is target lesion reduction and, instead, it is worth noting that progressive disease before observing a response is common both in patients treated with T-VEC alone or in combination with checkpoint inhibitors. This pattern of pseudo-progression has been widely described and reinforces the importance of continuing treatment beyond progression in the event of appearance of a new lesion or limited increase in existing ones [55,74].

In advanced melanoma, the combination of oncolytic viruses has been tested in clinical trials using T-VEC together with systemic administration of a checkpoint inhibitor: ipilimumab or pembrolizumab. A Phase 2 trial of T-VEC in combination with ipilimumab met its primary endpoint, resulting in a significantly higher ORR without additional safety signals than ipilimumab (39% vs. 18%, *p* = 0.002). Another Phase Ib trial tested the association of T-VEC and pembrolizumab with a confirmed ORR of 67% with a CR rate of 43%. In the coupled translational study, this association was able to increase CD8+ T cells, while PD-L1 and IFN-γ upregulation were observed in tumors from responders.

The positive effect of T-VEC, as a monotherapy or in combination with checkpoint inhibitors, which was observed in both injected and uninjected (including visceral) melanoma lesions, indicated that a systemic antitumor immune response was triggered. These results suggest that T-VEC may improve the efficacy of checkpoint inhibitor immunotherapies by changing the tumor microenvironment and support the rationale that combining immunotherapies with complementary mechanisms of action may yield augmented antitumor responses.

Interestingly, adverse events, both in monotherapy and in combination with immune checkpoints inhibitors are mild and easily reversible, leading to a new efficient and well tolerated treatment opportunity in those melanoma patients with injectable lesion and low tumor burden.

Trials are ongoing to confirm clinical results on larger number of patients and in comparison with the best standards of care, in order to confirm this approach is able to achieve high efficacy with low toxicity. Furthermore, new generation clinical trials incorporate regular sampling of both peripheral blood and tumor tissue, allowing basic and translational research, which will give insight on the mechanisms regulating tumor versus T-cells balance in the microenvironment and will characterize the immune response exploring its correlations with clinical outcomes. 

## Figures and Tables

**Figure 1 cancers-13-01383-f001:**
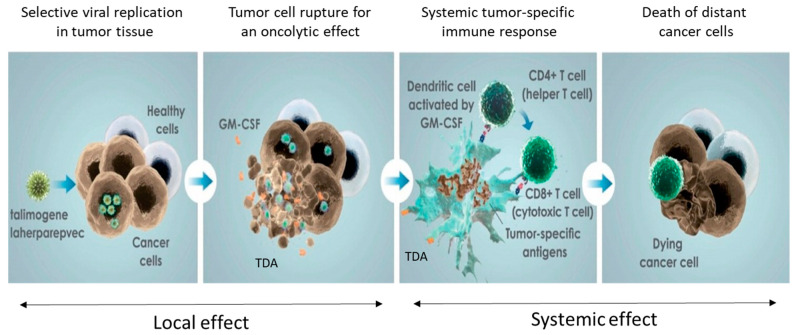
T-VEC proposed mechanism of action. TDA tumor-derived antigen. GM-CSF: Granulocyte–macrophage colony-stimulating factor. [33]. Image courtesy of Amgen Inc. Open access: used under the terms of the Creative Commons Attribution 4.0 International License (http://creativecommons.org/licenses/by/4.0/). Accessed on 22 January 2021.

**Figure 2 cancers-13-01383-f002:**
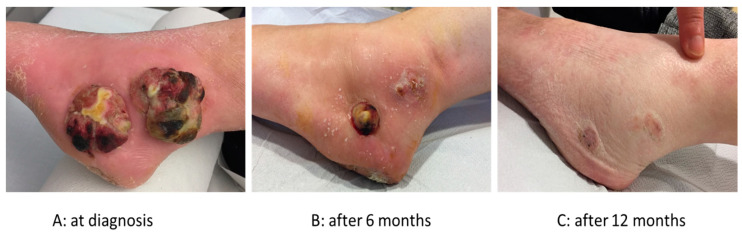
62 years old patient presenting with 2 huge malleolar lesions and lymphnodal metastases from acral melanoma. She was treated with T-VEC over a period of 2 years, achieving a complete remission, which is lasting after 2 years from the end of treatment. (**A**). At diagnosis; (**B**). after 6 months; (**C**). After 12 months.

**Table 1 cancers-13-01383-t001:** Genetic characteristics of T-VEC construct. GM-CSF: Granulocyte-macrophage colony-stimulating factor; HSV-1: Herpes simplex virus type 1. US11: Unique short glycoprotein 11.

T-VEC Genetic Components	Conferred Function
HSV-1 JS1	promotes preferential viral targeting of tumor cells
ICP34.5 gene deletion	Attenuates the natural viral pathogenicity allowing the virus to replicate selectively in tumor cells
ICP47 gene deletion	Reduces virally mediated suppression of antigen presentation and increase the expression of the HSV US11 gene
HSV US11 gene	Promotes virus growth in tumor cells without impairing tumor selectivity
GM-CSF cassette insertion	GM-CSF expression increases the recruitment of antigen-presenting cells and triggers a systemic antitumor immune response

**Table 2 cancers-13-01383-t002:** Concluded and ongoing clinical trials using T-VEC in monotherapy or combination. DOR: durable response rate, ORR: overall response rate, CR: complete remission, PFS: progression free survival, OS: overall survival.

Intratumoral Therapy	Performed Studies on Melanoma: Results	Ongoing Studies on Melanoma
Talimogen Laherparepvec (T-VEC)	Phase III (OPTiM), Efficacy: DOR 16.3%, ORR 26.4%, CR 10.8%. Safety: Well tolerated, with most common AEs being fatigue, chills, and pyrexia. Phase II (T-VEC + ipilimumab) Efficacy: 39 ORR %, CR 13%. Safety: Well tolerated, with most common AEs being fatigue, chills, and pyrexia. Phase Ib T-VEC + pembrolizumab (Masterkey-265) Efficacy: CR 43%, 4y PFS 56%, OS: 71%. Safety: Well tolerated, with most common AEs being fatigue, chills, and pyrexia	Phase III study of Pembrolizumab +/− T-VEC (KEYNOTE-034) (NCT02263508) Single-arm study to evaluate the role of the immune response to T-VEC, (NCT02366195) Phase II single arm study of T-VEC/pembrolizumab in patients refractory to anti–PD-1/PD-L1, (NCT02965716) Phase II study of surgery +/− T-VEC in the neoadjuvant setting, (NCT02211131)

## Data Availability

No new data were created or analyzed in this study. Data sharing is not applicable to this article.

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
