# Peer review of "Talimogene Laherparepvec (T-VEC): An Intralesional Cancer Immunotherapy for Advanced Melanoma"

_cancers, 2021, doi:10.3390/cancers13061383_

Round 1

Reviewer 1 Report

The authors have provided a detailed review on the clinical studies of T-VEC for patients with advanced melanoma, either alone or in combinations.

There have been quite a few review articles on this first-in-the class anti-cancer drug in the last few years. This one has provided comprehensive and up-to-date information. Therefore, it is a useful review article.

Minor points are,

  1. Table 2. The second column “Proposed mechanism of action” is not needed here. The mentioned 4 mechanisms are common knowledge for qualified readers of this article. The contents should be stated in the Introduction and in the introductory paragraph of the combination studies.
  2. Some typos throughout the manuscript. (1). Line 174. “were” should be “when”? (2). Line 176. “TVEC” should be “T-VEC”.
  3. References: The list has been complied with mixed formats and a number of references missed some of the key information, and with other information repeated twice.  This could be corrected easily if the authors use one of the reference management software.  For the missing information: For example, (1). Ref #33. The title of the article; (2). Ref #44. The official publication year is 2018, not 2017. The full information is, J Clin Oncol, 2018; 36(17):1658-1667.   (3). Ref #69. No time of publication (year).  (4). In a number of references, the year of publication has been stated repeatedly.

Author Response

Thank you for the comments and suggestions.

We have modified the manuscript accordingly, by eliminating the proposed mechanisms of action in table 2, which has been included in the Abstract because a dedicated section is explaining it in detail. We also reviewed the typos errors throughout the manuscript and we corrected the reference list by homogenizing the information of the reported publications as per Cancers format.

Thank you

Reviewer 2 Report

Page 1 Abstract line 28:  should be stated that the comparator GM-CSF was injected subcutaneously daily for two weeks and not intratumor

Page 2 line 49-50 intent of the English verbiage is unclear.

Line 51-53 English translation not totally clearer

Page line 109 suggest saying…traditional pharmacokinetic studies were not relevant and therefore not performed.

Page 4 line 151 Should specifically not % that were in each stage of IIIb, IIIc, IV M1a, IV M1b, and IV M1c.

Page 4 line 154 and other places.  Should reference and include content from more recent 2019 Andtbacka publication in J Immunother Cancer which is the final report of this study and has more complete information.

Page 5 line 190.  Need to state antiCTLA-4 checkpoint inhibitor

Page 5 line 195.  Assume intended to write “alone” rather than “one” next to p value.

Somewhere authors should comment on adoption intralesional injection of visceral and internal lesions via interventional radiology as opposed to being used primarily by direct injection by surgeons and medical oncologists.  Andtbacka and Kaufman are both surgeons.  What has been the adoption of this approach by medical oncologists?

Author Response

Thank you for reviewer's comments and suggestions.

We revised the whole manuscript following reviewer indications. In particular, we stated in the abstract that the comparator GM-CSF was injected subcutaneously daily for two weeks; we rewrite the unclear sentences line 49-50 and 51-53; we changed the sentence on pharmacokinetic studies; we modified the whole section related to the Optim trial including data from the final analysis published on JITC by Andtbacka and colleagues; we stated antiCTLA-4 checkpoint inhibitor and corrected typos as requested.

On page 5 line 195 there is not an error because "one" is referred to the arm and not to Ipilimumab, so the statement was not changed.

Finally, we also include in the Conclusions, a statement on the multidisciplinary approach, which is needed to perform this kind of treatment, including the emerging and extending role of interventional radiology.

Thank you